Unique antimicrobial activity in honey from the Australian honeypot ant (Camponotus inflatus)

http://orcid.org/0000-0001-8797-2643 Dong Andrew Z. 1
Cokcetin Nural 2
http://orcid.org/0000-0002-4638-0059 Carter Dee A. 1 3 dee.carter@sydney.edu.au
Fernandes Kenya E. 1
1 School of Life and Environmental Sciences, University of Sydney , Camperdown, NSW , Australia
2 Australian Institute for Microbiology and Infection, University of Technology , Sydney, NSW , Australia
3 Sydney Institute for Infectious Diseases, University of Sydney , Camperdown, NSW , Australia
Kumar Ravinder
Electronic publication date: 2023 Jul 26
Publication date: 2023
Volume: 11
Electronic Location ID: e15645
Received 2023 Apr 26; Accepted 2023 Jun 5
Copyright: © 2023 Dong et al.
Copyright year: 2023
Copyright holder: Dong et al.
License: This is an open access article distributed under the terms of the Creative Commons Attribution License, which permits unrestricted use, distribution, reproduction and adaptation in any medium and for any purpose provided that it is properly attributed. For attribution, the original author(s), title, publication source (PeerJ) and either DOI or URL of the article must be cited.
License URL: https://creativecommons.org/licenses/by/4.0/

Keywords: Honeypot ant, Honey, Antimicrobial activity, Functional food, Camponotus, Microbiome

Funding: The authors received no funding for this work.

==============================
Honey produced by the Australian honeypot ant (Camponotus inflatus) is valued nutritionally and medicinally by Indigenous peoples, but its antimicrobial activity has never been formally studied. Here, we determine the activity of honeypot ant honey (HPAH) against a panel of bacterial and fungal pathogens, investigate its chemical properties, and profile the bacterial and fungal microbiome of the honeypot ant for the first time. We found HPAH to have strong total activity against Staphylococcus aureus but not against other bacteria, and strong non-peroxide activity against Cryptococcus and Aspergillus sp. When compared with therapeutic-grade jarrah and manuka honey produced by honey bees, we found HPAH to have a markedly different antimicrobial activity and chemical properties, suggesting HPAH has a unique mode of antimicrobial action. We found the bacterial microbiome of honeypot ants to be dominated by the known endosymbiont genus Candidatus Blochmannia (99.75%), and the fungal microbiome to be dominated by the plant-associated genus Neocelosporium (92.77%). This study demonstrates that HPAH has unique antimicrobial characteristics that validate its therapeutic use by Indigenous peoples and may provide a lead for the discovery of novel antimicrobial compounds.

Introduction

Honey has been utilised since ancient times as a traditional remedy against various ailments. In recent years, there has been a resurgence of interest in the use of natural products such as honey as antimicrobials, in large part due to the growing crisis of antimicrobial resistance. While the vast majority of honey in the world is produced by the European honey bee Apis mellifera, many other insects collect, process, and store nectar in the form of honey (Crane, 1991). This includes stingless bees (Rosli et al., 2020), bumblebees (Bombus sp.) (Svanberg & Berggren, 2018), the Mexican honey wasp Brachygastra mellifica (Brock, Cini & Sumner, 2021), and various honeypot ant species (Andersen, 2002; Conway, 2003). One such example is the Australian honeypot ant, Camponotus inflatus. As a rare source of natural sugar in an arid environment, honeypot ants are highly prized as a bush food by Indigenous Australians and have a long history of nutritional and cultural significance (Islam et al., 2022). The Honey Ant Dreaming site is located in Central Australia and is shared by all Indigenous groups in the area (Jurra, 2000). For these groups, the honeypot ant represents their Dreaming or Tjukurpa, the Aboriginal philosophy based on the spiritual interrelation of people and things. In addition to their use as a food source, there are records of honeypot ant honey being used to treat sore throats and colds (Faast & Weinstein, 2020).

Honeypot ants are found only in environments that have an arid, dry, or desert-chaparral terrain. There are at least six different genera that live around the world and these have undergone convergent evolution and independently developed the same adaptation for novel nectar storage (Conway, 1991). Designated worker ants of the sterile helper caste that store food for the colony are known as “repletes”. These repletes are fed by other workers until their abdomens become engorged and semi-transparent (Froggatt, 1896). Becoming largely immobile, repletes take up a sacrificial role as a “living pantry” hanging off the roofs of their nests. Through antennae communication, repletes regurgitate this stored food during times of scarcity, which is then distributed via worker ants to the rest of the colony (Duncan & Lighton, 1994).

The antimicrobial activity of honeypot ant honey has not been studied, unlike that of honey bee honey where activity is attributed to physical characteristics such as high osmolarity and low pH, as well as other chemical factors. These chemical factors are highly variable and include plant-derived components such as flavonoids, amino acids, minerals, and phenolic acids (Almasaudi, 2021), and entomological additions such as glucose oxidase, which catalyses the production of hydrogen peroxide (H2O2), and antimicrobial peptides including defensin-1 and jellein-1, 2, and 4 (Brudzynski & Sjaarda, 2015). The total activity (TA) of a honey refers to the broad-spectrum activity that results from the synergistic efforts of these factors combined. Honey from different sources can have vastly differing activity levels and mechanisms of action but are broadly categorised into having either peroxide-activity (PA) or non-peroxide activity (NPA). For example, jarrah (Eucalyptus marginata) honey typically possesses high levels of H2O2 making it a PA honey, while manuka (Leptospermum scoparium) honey typically contains high levels of methylglyoxal (MGO) and retains its bioactivity even when H2O2 is removed, making it a NPA honey.

Honeypot ants worldwide are reported to source nectar from a variety of floral sources depending on availability and seasonality (Hölldobler, 1981). In Australia, Camponotus inflatus is thought to have a preferential association with mulga trees and the aphids that live on them, though they are reported to gather nectar from a variety of other floral sources at different times of the year including black corkwood and native fuchsia flowers (Conway, 1991; Islam et al., 2022). Mulga trees possess nectar-secreting plant glands known as extrafloral nectaries that attract honeypot ants, who in turn protect the plant against herbivores (Buckley, 1982). Aphids feed on the sugary mulga sap, metabolising its amino acids and honeypot ants stroke the aphids with their antennae, coaxing them to excrete excess honeydew from their anuses, which they then collect (Blüthgen & Feldhaar, 2010). In return, the ants provide hygienic services and protect the aphids from predators and parasitoids (Ness, Mooney & Lach, 2010).

Given the resurgence of interest in the medicinal value of honey bee honey, it is of interest to investigate the bioactivity of honey from other species, particularly one that has been utilised medicinally by Indigenous peoples for thousands of years. With little currently known about how honeypot ants process their honey, profiling their microbiome may provide insights into their diet, nutrient processing and metabolic capabilities, and the coevolutionary dynamics that might impact the properties of their honey. In this study, we determine the antimicrobial activity of Australian HPAH against a variety of bacterial and fungal pathogens, compare its physical and chemical properties with therapeutic-grade honey bee honey in order to determine potential mechanisms of action, and investigate the honeypot ant microbiome through metagenomic analysis. We report here for the first time the bacterial and fungal microbiome of Australian honeypot ants and the antimicrobial activity profile of their honey.

Materials and Methods

Sample collection

Honeypot ants and honeypot ant honey (HPAH) samples were collected from Kurnalpi, located in the Goldfields-Esperance region of Western Australia, on 22nd May 2022 with the help of local Indigenous guides. A Camponotus inflatus nest was located by searching for Mulga (Acacia aneura) trees in the area and then identifying a worker ant that would lead to the entrance of the nest. Careful excavation of the nest took place from 1 pm to 3 pm, exposing underground galleries containing repletes. These were surface sterilised and subsequently euthanised using 80% ethanol. HPAH was harvested by pricking the abdomen of a replete with a sterile needle and squeezing its contents. Ant bodies and honey samples were stored in the dark at 4 °C until use.

Honey sample preparation

HPAH was mixed thoroughly by pipetting, diluted to the target concentration in sterile water, and vortexed thoroughly prior to use. Honeys with known activity levels and marketed as therapeutically active were Barnes TA 10+ Jarrah honey (hereafter referred to as jarrah honey), which has peroxide-based activity, and Comvita UMF 18+ Manuka honey (hereafter referred to as manuka honey), which has methylglyoxal (MGO)-based non-peroxide activity. Artificial honey (1.5 g sucrose, 7.5 g maltose, 40.5 g fructose, 33.5 g glucose, 17 mL sterile water), was included as an inactive, non-floral, non-bee control. These honey samples were mixed thoroughly with a spatula, incubated at 35 °C for 30 min to dissolve crystals, diluted to the target concentration in sterile water, and vortexed thoroughly before use.

Antimicrobial susceptibility testing

A diverse range of pathogens were chosen for antimicrobial susceptibility testing including Gram-negative and Gram-positive bacteria, yeasts, and moulds. Bacterial strains, yeast strains, and mould strains excluding M. gypseum were maintained as glycerol stocks at −80 °C. Bacterial strains were grown on Nutrient Agar (NA; Oxoid) and incubated at 30 °C for 24 h before use. Yeast strains and mould strains excluding M. gypseum were grown on Potato Dextrose Agar (PDA; Oxoid) and incubated at 30 °C for 24–48 h. M. gypseum was maintained on an agar slope, grown on Oatmeal Agar (Sigma Aldrich St. Louis, MO, USA), and incubated at 30 °C for up to 7 days until good sporulation was obtained. The phenol equivalence (PE) assay was performed according to the method outlined in Irish, Blair & Carter (2011). This assay is the current industry standard for quantifying antimicrobial activity in honey and determines the activity of honey against Staphylococcus aureus in relation to phenol standards (% PE), with a greater number indicating more active honey. Antimicrobial susceptibility testing by broth microdilution in 96-well plates was performed in accordance with Clinical and Laboratory Standards Institute (CLSI) guidelines for aerobic bacteria M07-A10 (Clinical and Laboratory Standards Institute (CLSI), 2002b), yeasts M27-A4 (Clinical and Laboratory Standards Institute (CLSI), 2017), and filamentous fungi M38-A3 (Clinical and Laboratory Standards Institute (CLSI), 2002a). Broth microdilution assays determine the minimum inhibitory concentration (MIC), the lowest percentage of honey diluted in water that inhibits a certain amount of growth, with a smaller number indicating more active honey. Honeys were assayed at doubling dilutions beginning at 32% (w/v) and were diluted in either sterile water for total activity or freshly prepared 5,600 U/mL catalase solution for non-peroxide activity. Absorbance values at 600 nm relative to a growth control were used to determine the MIC100 (100% growth inhibition), MIC80 (80% growth inhibition) and MIC50 (50% growth inhibition). For heat treatments, honey was heated to 90 °C for 10 min using a heat block, before being allowed to return to room temperature naturally. Raw data is presented in Table S1.

Assessment of honey colour, pH, water content, and water activity

The optical density of honey samples at 50% (w/v) was measured at 450, 635, and 720 nm using a UV/Vis spectrophotometer (UV-1600PC; VWR International, Radnor, PA, USA) with sterile water as a blank. Colour intensity was calculated using the equation (A720−A450)×1000 and expressed in milli-absorbance units (mAU). Pfund value was calculated using the equation −38.70+371.39(A635) and expressed in mm. For pH measurements, 1 g of honey was diluted in 7.5 mL of sterile water and pH was determined using a pH meter (Seven Compact S220; Mettler Toledo, Greifensee, Switzerland). Brix value and moisture content were measured at 20 °C using a refractometer (HI96801; Hanna Instruments, Smithfield, RI, USA) according to the AOAC Official Method 969.38 (AOAC, 2023). Water activity (aw) was assessed using a water activity analyser (PRE; Aqualab Scientific, Pullman, WA, USA) at 25 °C with a correction of ±0.005 aw made per 0.1 °C deviation (Stoloff, 1978). Raw data is presented in Table S1.

FC and FBBB phenolics assays

For the Folin-Ciocalteu (FC) assay, 20 µL aliquots of 20% (w/v) honey samples were added to the wells of a 96-well plate in triplicate. To each well, 100 µL of FC reagent (1 mL Folin-Ciocalteu reagent in 30 mL sterile water) was added and the plate was incubated at room temperature for 5 min in the dark. Next, 80 µL of Na2CO3 solution was added with incubation at room temperature for 2 h in the dark. Absorbance was measured at 760 nm using a microplate reader (CLARIOstar Plus; BMG Labtech, Ortenberg, Germany). For the Fast Blue BB (FBBB) assay, 200 µL aliquots of 20% (w/v) honey samples were added to the wells of a 96-well plate in triplicate. To each well, 20 µL of 0.1% Fast Blue BB reagent was added and thoroughly mixed by pipetting up and down 50 times. Next, 20 µL of 5% NaOH solution was added, and the plate was incubated at room temperature for 45 min in the dark. Absorbance was measured at 420 nm using a microplate reader. For both assays, gallic acid standards ranging from 0.06–0.18 mg/mL were used to generate a standard curve and the resulting equation for the line of best fit was used to calculate the phenolics content of honey samples, expressed as mg of gallic acid equivalent per kg of honey (mg GAE/kg). Raw data is presented in Table S1.

FRAP and DPPH antioxidant assays

For the ferric-reducing antioxidant power (FRAP) assay, FRAP reagent consisting of 1:1:10 (v/v/v) of 10 mM TPTZ in 40 mM HCl, 20 mM FeCl3, and 300 mM pH 3.6 acetate buffer was prepared fresh and incubated at 37 °C prior to use. Honey samples (20 µl of 20% (w/v)) were added to the wells of a 96-well plate in triplicate. Next, 180 µL of FRAP reagent was added and plates were incubated at 37 °C for 30 min. Absorbance was measured at 594 nm using a microplate reader. Iron (II) sulfate (FeSO4) standards ranging from 200–1,200 µM, made freshly and stored on ice until use, were used to generate a standard curve and the resulting equation from the line of best fit was used to calculate FRAP value, expressed as µmol Fe2+/kg. For the 2,2-diphenyl-1-picrylhyrazyl (DPPH) assay, 10 µL aliquots of 20% (w/v) honey samples were added to the wells of a 96-well plate in triplicate. Next, 100 µL of 100 mM pH 5.5 sodium acetate buffer and 250 µL of DPPH reagent (130 µM DPPH in methanol) were added with incubation at room temperature for 2 h in the dark. Absorbance was measured at 520 nm using a microplate reader using methanol as a blank. Trolox standards at pH 7 ranging from 100–600 µM were used to generate a standard curve and the resulting equation from the line of best fit was used to calculate radical scavenging activity, expressed as µmol Trolox equivalent per kg of honey (µmol TE/kg). Raw data is presented in Table S1.

HRP hydrogen peroxide assay

The horseradish peroxidase (HRP) assay was performed according to the method outlined in Lehmann et al. (2019) with minor modifications. Briefly, honey samples were diluted to 50% (w/v) with sterile water, passed through a 0.22 µm pore filter, and 1 mL was aliquoted in six well-plates to allow for adequate overhead aeration. Samples were further diluted to 25% (w/v) with either sterile water or 5,600 U/mL catalase solution and incubated at 35 °C with 180 rpm shaking in the dark. At each timepoint, 40 µL aliquots of each sample were taken, 135 µL of freshly prepared HRP reagent (50 µg/mL o-dianisidine and 20 µg/mL HRP in 10 mM pH 6.5 sodium phosphate buffer) was added, and samples were incubated at room temperature for 5 min in the dark before the reaction was terminated by the addition of 120 µL of 6 M H2SO4. Absorbance was measured at 550 nm using a microplate reader (ELx800; BioTek Instruments, Winooski, VT, USA). Honey blanks were taken at each timepoint by adding 135 µL of sodium phosphate buffer in place of HRP reagent. H2O2 standards ranging from 0.5–1,024 µM were used to generate a standard curve and the resulting equation from the line of best fit was used to calculate the amount of H2O2 in each sample. Raw data is presented in Table S1.

DNA preparation

Seven ants ranging in size from 0.1 to 1.66 g were individually processed and analysed. Ant bodies were surface sterilised with 1% (v/v) bleach for 3 min and thoroughly rinsed with sterile water. Whole ants were placed individually in tubes containing 500 mg of 2 mm glass beads in 500 µL of PBS and homogenised in a beat beater (PowerLyzer 24; Qiagen, Hilden, Germany) using six cycles of 30 s at 3,000 rpm with 30 s rests between. The mixture was briefly centrifuged, and the supernatant transferred into fresh tubes. DNA was extracted using a DNeasy Blood and Tissue Kit (Qiagen, Hilden, Germany) following the manufacturer’s instructions for animal tissue.

PCR & gel electrophoresis

PCR and gel electrophoresis were conducted to confirm the presence of sufficient bacterial and fungal DNA in samples. The V3-V4 region of the 16S rRNA gene was amplified using the primer pair 341F/805R. The internal transcribed spacer 1 (ITS1) region was amplified using the primer pair ITS1F/ITS2. PCR conditions were as follows: initial denaturation at 94 °C for 30 s, followed by 25 cycles of denaturation at 94 °C for 30 s, annealing at 60 °C for 30 s, and extension at 68 °C for 90 s, then a final extension at 68 °C for 5 min. PCR products were analysed by electrophoresis on a 0.8% agarose gel in TAE buffer run at 90 V for 1 h.

Amplicon sequencing & analysis

DNA was sent to Ramaciotti Centre for Genomics at the University of New South Wales, Sydney for 16S V3-V4 amplicon sequencing with the 341F-805R primer set using the Illumina Miseq v3 2 bp × 300 bp platform, and to BGI Genomics, Hong Kong for ITS1 amplicon sequencing with the ITS1F-ITS2 primer set using the DNBSEQ PE300 platform. Raw sequence reads were processed in R v4.2.2 using the DADA2 pipeline. Default parameters were used to filter and trim, learn error rates, merge paired reads, and remove chimeras with the following adjustments: the truncLen parameter was adjusted to c(260, 220) to allow for sufficient overlap of forward and reverse reads for merging of the V3-V4 amplicons, and this step was not performed for the variable length ITS1 amplicons. Taxonomy was assigned using the SILVA database release 138.1 for 16S, and the UNITE database release 27.10.2022 for ITS. Non-bacteria, mitochondria and chloroplast were filtered out from 16S taxonomic tables and non-fungi from ITS taxonomic tables. Taxonomic relative abundances were calculated using the phyloseq R package. Raw metagenomic data obtained during this study is publicly available in the NCBI Sequence Read Archive under Bioproject ID PRJNA957126.

Results

Honeypot ant honey has activity against bacteria, yeasts, and moulds

The antimicrobial activity of honeypot ant honey (HPAH) was tested and compared against active peroxide (jarrah) and non-peroxide (manuka) based bee honeys. Using the PE assay, the total activity of HPAH against S. aureus was found to be 8.3% PE (Table 1). This is in the low activity range (5–10%), and was lower than the 11.6% PE total activity of the jarrah honey, which is in the ‘potentially beneficial for therapeutics’ range (10–20%), and the 19.7% PE total activity of the manuka honey which is approaching the high activity (>20%) range (Irish, Blair & Carter, 2011). HPAH and the jarrah honey had no detectable non-peroxide activity, while the manuka honey had non-peroxide activity of 20.1% PE. The artificial honey control had no detectable total or non-peroxide activity.

Table 1 Total and non-peroxide activity of honeypot ant and active bee honeys against Staphylococcus aureus determined by the phenol equivalence assay.

Honey sample	Phenol equivalence (%)	
Total activity	Non-peroxide activity	
Honeypot	8.3	<5	
Artificial	<5	<5	
Jarrah	11.6	<5	
Manuka	19.7	20.1	

Although it is the current industry standard, the PE assay only tests activity against a single organism, and as a diffusion-based assay can sometimes underestimate the activity of honey samples with unique properties (Hossain et al., 2022). To address these issues, broth microdilution assays were used to assess the total activity of HPAH against a range of pathogenic microbes including bacteria, yeasts, and moulds (Table 2). Artificial honey produced a MIC100 of >32% for all species tested, except P. aeruginosa which is more susceptible to osmolarity, with an MIC100 of 32%. Unlike in the PE assay, HPAH was found to be more active against S. aureus (MIC100 8%) than the jarrah honey (MIC100 16%) and was on par with the manuka honey (MIC100 8%). HPAH had very low detectable activity against the three other bacterial species tested, E. faecalis, P. aeruginosa, and E. coli, with an MIC100 of >32% and an MIC50 of 32% for all three. The jarrah honey had the same MIC100 of 16% for all four bacterial species. The manuka honey had an MIC100 of 16% for E. faecalis, P. aeruginosa, and E. coli.

Table 2 Total activity of honeypot ant and active bee honeys (% w/v) against various bacterial and fungal pathogens determined by broth microdilution.

Group	Species	Honeypot1	Artificial	Jarrah	Manuka	
MIC100	MIC80	MIC50	MIC100	
Bacteria	Staphylococcus aureus	8	8	8	>32	16	8	
Enterococcus faecalis	>32	32	32	>32	16	16	
Pseudomonas aeruginosa	>32	>32	32	32	16	16	
Escherichia coli	>32	32	32	>32	16	16	
Yeasts	Candida albicans	>32	>32	>32	>32	16	16	
Candida glabrata	>32	>32	>32	>32	16	16	
Cryptococcus neoformans	16	8	8	>32	16	16	
Cryptococcus deuterogattii	32	16	16	>32	16	16	
Moulds	Aspergillus fumigatus	16	16	16	>32	16	16	
Aspergillus flavus	16	16	8	>32	16	16	
Fusarium oxysporum	32	32	32	>32	16	8	
Microsporum gypseum	32	32	32	>32	8	8	
Note:

1 MIC100 = 100% inhibition, MIC80 = 80% inhibition, MIC50 = 50% inhibition.

For the yeast species, HPAH had no detectable activity against either Candida species (MIC50 > 32%), but was active against both Cryptococcus species with an MIC100 of 16% for C. neoformans and 32% for C. deuterogattii. The jarrah and manuka honeys had the same MIC100 for all yeast species at 16%. For moulds, HPAH was active against Aspergillus species with an MIC100 of 16% for both A. fumigatus and A. flavus, but was less active against F. oxysporum and M. gypseum, with an MIC100 of 32%. The inverse was seen for the jarrah and manuka honeys which were more effective against F. oxysporum (MIC100 8% and 16%, respectively) and M. gypseum (MIC100 8%) than A. fumigatus and A. flavus (MIC100 16%).

Non-peroxide components contribute to the activity of honeypot ant honey

To investigate potential active components, HPAH was subjected to catalase and heat treatment. Catalase degrades H2O2, while heat denatures glucose oxidase, the enzyme that catalyses the production of H2O2 from glucose and water. Both catalase and heat treated HPAH was tested via broth microdilution for bacterial and fungal species producing a MIC100 ≤ 32% (Table 3). For S. aureus, catalase treatment decreased the activity of HPAH, raising all MIC values from 8% to 32%. The MIC100 of heat-treated HPAH was the same as catalase-treated HPAH at 32%, while the MIC50–80 increased from 16% to 32% indicating a potential role of other non-glucose oxidase heat-labile components in the antimicrobial activity of HPAH. For the fungi, catalase-treatment increased the MIC100 of HPAH for C. neoformans from 16% to 32% but had no effect on the MIC100 of C. deuterogattii (32%), A. fumigatus (16%), or A. flavus (16%) indicating that the inhibition of these pathogens by HPAH is likely due to non-peroxide mechanisms alone. The limited volume of HPAH available meant heat treatment could not be assessed for the fungal species.

Table 3 Non-peroxide activity of honeypot ant honey (% w/v) after heat or catalase treatment determined by broth microdilution.

Species	No treatment1	Catalase treatment2	Heat treatment3	
MIC100	MIC80	MIC50	MIC100	MIC80	MIC50	MIC100	MIC80	MIC50	
Staphylococcus aureus	8	8	8	32	16	16	32	32	32	
Cryptococcus neoformans	16	8	8	32	16	16	–	–	–	
Cryptococcus deuterogattii	32	16	16	32	16	8	–	–	–	
Aspergillus fumigatus	16	16	16	16	16	16	–	–	–	
Aspergillus flavus	16	16	8	16	16	8	–	–	–	
Notes:

1 MIC100 = 100% inhibition, MIC80 = 80% inhibition, MIC50 = 50% inhibition.

2 For catalase treatment, samples were diluted with a 5,600 U/mL catalase solution.

3 For heat treatment, samples were heated at 90 °C for 10 min.

The horseradish peroxidase (HRP) assay was used to determine the amount and kinetics of H2O2 production in the different honey samples over 4 h. HPAH had low levels of H2O2, with a maximum of 2.1 µM detected 1 h into the assay (Fig 1; Table 4). This was lower than the manuka honey, with a maximum of 4.5 µM detected 1.5 h into the assay, and much lower than the jarrah honey that exhibited a typical ‘inverted U-shape’ curve with H2O2 peaking at 9.4 µM at 1 h into the assay. The artificial honey control had no detectable H2O2 production.

Figure 1 Honeypot ant honey produces low amounts of hydrogen peroxide.

Hydrogen peroxide (H2O2) production by honeypot ant and active bee honeys measured using the horseradish-peroxidase assay over the course of 4 h.

Table 4 Chemical properties of honeypot ant, jarrah, and manuka honeys.

Property	Honeypot	Artificial	Jarrah	Manuka	
Maximum H2O2 (µM)	2.1	0	9.4	4.5	
Time at Maximum H2O2 (h)	1	N/A	1	1.5	
Sugar content (°Brix)	63.2	78.1	83.1	79.4	
Moisture content (%)	36.5	21.5	16.5	20.2	
Water activity (aw)	0.80	0.58	0.54	0.60	
pH	3.4	4.5	4.5	3.8	
Colour intensity (mAU)	1,844	50	1,657	2,433	
Colour (Pfund value)	165	0	203	248	
Colour (Pfund colour)	Dark amber	Water white	Dark amber	Dark amber	
Phenolics via FC (mg GAE/kg)	437	38	471	558	
Phenolics via FBBB (mg GAE/kg)	159	0	295	434	
Antioxidants via FRAP (µmol Fe2+/kg)	3,268	0	4,158	4,468	
Antioxidants via DPPH (µmol TE/kg)	4,498	1,246	5,098	5,476	

Honeypot ant honey has very different properties to jarrah and manuka honey bee honeys

Various chemical properties of HPAH that may contribute to antimicrobial activity were measured and compared to the manuka and jarrah honeys (Table 4). The moisture content of HPAH at 36.5% was considerably higher than manuka (20.2%) or jarrah (16.5%), and the sugar content of HPAH at 63.2° Brix was considerably lower (manuka 79.4° Brix; jarrah 83.1° Brix). Water activity (aw), which is the measure of unbound or biologically available water, was also substantially higher in HPAH at 0.80 than in manuka (0.60) or jarrah (0.54). The pH of HPAH at 3.4 was lower than manuka (3.8) and jarrah (4.5), but within what is considered a normal range for honey bee honey (~3.2 to 4.5). The colour intensity of HPAH was 1,844 mAU with a Pfund value of 165 mm, placing it in dark amber, the darkest colour category, along with the jarrah and manuka honeys.

Total phenolics content was assessed using the Folin-Ciocalteu (FC) assay, which works via a redox reaction and is thus affected by non-phenolic reducing compounds, and the Fast Blue BB (FBBB) assay, which is more specific and based on a direct reaction with active hydroxyl groups in the phenolic compounds. Total phenolics detected by FBBB (range 159–434 mg GAE/kg) were consistently lower than detected by FC (range 437–568 mg GAE/kg), confirming an interference by non-phenolic compounds in the FC assay. Nonetheless, the trends of each sample remained the same, with HPAH (FC = 437 mg GAE/kg; FBBB = 159 mg GAE/kg) lower than the jarrah honey (FC = 471 mg GAE/kg; FBBB = 295 mg GAE/kg), which was in turn lower than the manuka honey (FC = 558 GAE/kg; FBBB = 434 mg GAE/kg).

Antioxidant activity was assessed using the ferric reducing antioxidant power (FRAP) assay, which measures the capacity of samples to reduce Fe3+ to Fe2+, and the 2, 2-diphenyl-1-picrylhydrazyl (DPPH) assay, which measures the ability of samples to scavenge the DPPH free radical. Antioxidant activity detected by FRAP (range 0–4,468 µmol Fe2+/kg) was consistently lower than measured by DPPH (range 1,246–5,476 µmol TE/kg) although the trends of each sample remained the same, showing good correlation between assays. The antioxidant activity of HPAH was 3,268 µmol Fe2+/kg via FRAP and 4,498 µmol TE/kg via DPPH, placing it below jarrah (4,158 and 5,098, respectively) and manuka (4,468 and 5,476, respectively) honey.

The bacterial and fungal microbiomes of honeypot ants are each dominated by a single genus

16S and ITS rRNA gene sequencing were used to assess the bacterial and fungal composition of the honeypot ant microbiome, respectively. Seven repletes were chosen, with various levels of honey engorgement, from the smallest weighing 0.1 g to the largest weighing more than 16× greater at 1.66 g (Fig 2). DNA was extracted from individual whole honeypot ant bodies. The bacterial microbiome of the honeypot ant samples was almost exclusively Candidatus Blochmannia (99.75%), a known endosymbiont of the Camponotus genus. Within the remaining 0.14% of ASVs that were identifiable, the next 10 most abundant genera were as follows: Gilliamella, Pseudomonas, Enterobacter, Bacillus, Apilactobacillus, Erwinia, Cutibacterium, Gaiella, Actinomyces, and Bombella. The fungal microbiome was also dominated almost entirely by a single taxon, Neocelosporium (92.77%), with the second most abundant genera being Endosporium (5.51%). Within the remaining 0.21% of ASVs that were identifiable, the genera were as follows: Penicillium, Spirographa, Aureobasidium, and Metapochonia. Phylogenetic trees showing all taxa identified at the level of genus are presented in Fig. 3.

Figure 2 The bacterial and fungal microbiomes of the honeypot ant are both dominated by individual species.

(A) The appearance and weights of honeypot ant repletes selected for microbiome analysis at various levels of honey engorgement. (B) Relative abundance of bacterial and fungal genera in the honeypot ant microbiome, averaged across seven ants for bacteria and four ants for fungi. Charts on the right show relative abundance of genera within the “Other” category shown on left charts.

Figure 3 Phylogenetic trees of the bacterial and fungal honeypot ant microbiome at genus level.

Trees were generated using Geneious (6.0.6) and visualised with iTOL (v6). Genera are colour-coded at phylum level.

To investigate whether HPAH contained a microbial composition itself, honey was spread onto nutrient agar, potato dextrose, and oatmeal agar plates and incubated at 20 °C or 35 °C to provide conditions that would be suitable for growth of a variety of bacteria and fungi (Fig. 4). HPAH was found to be relatively sterile compared to raw bee honey which can harbour a wide variety of microbes (Sereia et al., 2017). After 7 days of incubation, a single colony was observed on potato dextrose agar at 20 °C and on oatmeal agar at 35 °C. After 21 days of incubation, further growth was seen on nutrient and potato dextrose agar at 20 °C, and oatmeal agar at 35 °C. Plates with visible growth were scraped and DNA extracted for 16S and ITS rRNA gene sequencing to identify colonies. Analysis of the plates identified the following bacterial genera: Stenotrophomonas (32.09%), Rummeliibacillus (29.71%), Bacillus (26.86%), Paenibacillus (9.11%), and Streptomyces (2.16%). Fungal genera identified were Neocelosporium (93.75%), and Endosporium (6.25%).

Figure 4 Honeypot ant honey contains few microbes.

(A) Spread plates of honeypot ant honey on nutrient, potato dextrose, and oatmeal agar incubated at 20 °C or 35 °C for 7 and 21 days. (B) After 21 days, plates with visible growth were scraped, and DNA was extracted and analysed via 16S or ITS1 rRNA gene sequencing to determine the relative abundance of bacterial and fungal genera present.

Discussion

Medicinal honey has gained much attention as an effective broad-spectrum antimicrobial, though little investigation has been undertaken on honey produced by insects other than the honey bee. In this study, we investigate the honey produced by the honeypot ant Camponotus inflatus, which has a long history of cultural importance to the Indigenous people of Australia as a delicacy and a bush medicine. The honeypot ant, like the honey bee, is a eusocial hymenopteran, and both collect nectar to produce honey for long-term storage in their respective colonies. However, differing sources of nectar, the unique form of storage and chemical composition, and unknown entomological additions in HPAH make the properties of the end-product quite different from honey bee honey (summarised in Fig. 5).

Figure 5 Characteristics of honey bee (Apis mellifera) honey and honeypot ant (Camponotus inflatus) honey that contribute to their antimicrobial properties.

Honeys produced by honey bees and honeypot ants are influenced by their specific forage sources, entomological additions and storage types, resulting in particular, unique characteristics.

Honeypot ant honey has unique species-specific and non-peroxide antimicrobial activity that may reflect evolutionary pressures

Comparing the activity of HPAH against therapeutic-grade active jarrah and manuka honeys, we found a markedly different activity profile, with HPAH outperforming these honeys against some pathogens but exhibiting low or no activity against others. Additionally, we found strong evidence for non-peroxide mechanisms of action, with HPAH producing low levels of H2O2 and retaining activity against S. aureus, Cryptococcus sp., and Aspergillus sp. after catalase or heat treatment. While the bee honeys had similar activity against all bacterial species tested, HPAH had high activity against S. aureus (MIC100 8%) and low activity against the others (MIC50 32%). These results suggest that the activity of HPAH against S. aureus is most likely due to a non-peroxide component, with the low level of activity against the other bacteria due to H2O2. The large and species-specific differences in activity against fungal organisms, and the broader range of antimicrobial activity of HPAH (MIC100 8–>32%) compared to the bee honeys (MIC100 8–16%) further suggest the presence of non-peroxide compounds. While peroxide clearly plays a role in the activity against certain species, with activity substantially diminished after the addition of catalase, the relatively small amounts of H2O2 produced by HPAH suggests that it would not be sufficient on its own to account for the activity observed. It is thus likely that one or more non-peroxide components in HPAH synergise with H2O2 to exert antimicrobial activity but are less or non-functional without it. Overall, these results indicate that HPAH has unique underlying mechanisms of action that are derived from the honeypot ant.

A notable difference of HPAH compared to the bee honeys was the substantial variation in susceptibility across different bacterial and fungal species. Evolutionary pressure exerted by the honeypot ant environment may be responsible for this variable activity, particularly toward the different fungal species. Honeypot ants live exclusively in dry, arid, or desert-like environments. Both A. fumigatus and A. flavus are ubiquitous thermotolerant fungi with an ecological niche in soil debris (Bhabhra & Askew, 2005; Latgé, 1999). A study surveying various plots of Australian soil found A. fumigatus to be present in 79% of all plots across different climate regions, and in 100% of plots in hot, arid, desert climates (Ellis & Keane, 1981). This makes it likely that honeypot ants have evolved to be resistant to Aspergillus sp. for their survival, reflected in the strong activity (MIC100 16%) of HPAH against Aspergillus sp. HPAH had similarly strong activity (MIC100 16%) against Cryptococcus sp., which are environmental saprotrophs that thrive on decaying wood and soil. Cryptococcus sp. are spread among vegetation by a variety of animal species and insects including ants (Edwards et al., 2021), making it likely that worker ants come into contact with this pathogen as they venture through trees and other plants in search of food. Conversely, HPAH was not very active (MIC100 32%) against Microsporum gypseum, a member of the dermatophyte group of fungi, which tend to be highly susceptible to active bee honeys. Although M. gypseum is geophilic and found in soils worldwide (Souza et al., 2016), soil with less than 5% moisture does not support its growth making it mostly found in gardens, parks and soils protected by shade (Ranganathan & Balajee, 2000). This makes it unlikely that M. gypseum would be present in the honeypot ant environment.

Few taxa dominate the bacterial and fungal microbiome of the honeypot ant

Over 99% of the bacterial microbiome of the honeypot ant was comprised of a single genus, Candidatus Blochmannia (Blochmannia). This is a mutualist that has been found in all Camponotus species screened to date (de Souza et al., 2009; Degnan et al., 2004; Feldhaar et al., 2007; Sauer et al., 2002; Schröder et al., 1996). Blochmannia generally live in specialised bacteriocyte cells located in the midgut, however they have also been found in the crop and the hindgut suggesting that they may have the capacity to invade other gut tissues (He, Wei & Wheeler, 2014). The vertical maternal transmission of Blochmannia to ant offspring suggests that it is engaged in a long-term stable relationship with its host; the hallmark of a primary endosymbiont (Degnan et al., 2004). These highly developed symbiotic systems are often found in insects that specialise on unbalanced diets (Sauer et al., 2000). With honeypot ants feeding largely on honeydew and sugary secretions, Blochmannia is likely involved in essential metabolic processes, such as nitrogenous compound recycling, that allow the honeypot ant to occupy its ecological niche (Zientz, Dandekar & Gross, 2004). Endosymbionts are often also involved in modulation of the host immune system, priming it for more efficient protection against pathogens. However, studies investigating this possibility in other Camponotus species have reported mixed findings, ranging from positive effects with increased Blochmannia numbers in C. fellah (de Souza et al., 2009), to neutral (Sauer et al., 2002) or negative (Sinotte et al., 2018) effects in C. floridanus. Other bacterial taxa in the honeypot ant microbiome, although comprising a very small percentage, included several genera identified in the core honey bee microbiome including Gilliamella, Apilactobacillus, and Bombella. These are thought play a role in digestion by secreting substances that aid in the metabolism of certain toxic carbohydrates in the nectar diet (Ahmad et al., 2022; Härer, Hilgarth & Ehrmann, 2022; Zheng et al., 2016).

The fungal microbiome has not previously been profiled in any Camponotus species. Neocelosporium, the dominant fungal genus, is an environmental saprotroph involved in nutrient recycling and produces spreading mycelia on leaves. This genus was first identified in 2018 and to date contains only two species: Neocelosporium eucalypti isolated from Eucalyptus cyanophylla trees in southern Australia (Crous et al., 2018), and Neocelosporium corymbiae isolated from Corymbia variegata trees in eastern Australia (Crous et al., 2021). It is not known if Neocelosporium is associated with leaves of the mulga tree or with other plants likely to be in the vicinity of honeypot ant nests, or if it plays a role in the biology or nutrition of Camponotus, and these would be interesting areas for further study. The five remaining genera identified in the fungal microbiome all also have environmental niches in plants, water, and/or the soil suggesting that the honeypot ant fungal microbiome is largely obtained from foraging and environmental exposure (Bizarria, Pagnocca & Rodrigues, 2022; Flakus et al., 2019; Siedlecki et al., 2021; Tsuneda et al., 2008). Unsurprisingly, the microbiome of HPAH was largely sterile and did not contain any yeasts. As the honey is stored within the ant’s body, any fermentation or spoilage by transient microbes would likely kill the ant. The mechanism by which HPAH is rendered sterile is unknown but may include physical filtration of small particles by the infrabuccal pocket, which has been identified in other ant species but never studied in C. inflatus (Zheng et al., 2022), or acidopore grooming by which acidic poison gland secretions are swallowed for microbial control, which has likewise been identified in other Camponotus species but not studied in C. inflatus (Tragust et al., 2020).

Potential sources of active compounds in honeypot ant honey

In medicinal bee honey, non-peroxide activity is valued in a clinical setting due to its comparative resistance to heat, light and catalase (Cooper, Molan & Harding, 1999). The known suite of non-peroxide factors in honey includes phenolic compounds and phytochemicals that are heavily influenced by nectar source (Johnston et al., 2018), and proteins and antimicrobial peptides that are derived from the bee. The bee honey with the closest nectar source to HPAH may be honeydew honey, derived from the excretions of plant sucking insects such as aphids, rather than from plant nectar (Codex Alimentarius, 2001). Honeydew honey has a different sugar profile from most other honeys, and is usually darker in colour, with greater phenolic and antioxidant content than blossom honeys (Pita-Calvo & Vázquez, 2017). In a study looking at the antimicrobial activity of honeydew honey, Bucekova et al. (2018) suggested there may be non-peroxide components that synergise with H2O2, similar to what we propose for HPAH. However, they reported a much higher accumulation of H2O2 and inhibition of P. aeruginosa, which was not seen in our study. Entomological differences as well as differences in honeydew foraged by bees compared to mulga-derived honeydew collected by the ants may underlie differences in the resulting properties of HPAH, suggesting it is unique among honey types.

Likely entomological candidates for the non-peroxide components of HPAH are antimicrobial peptides (AMPs). These are short proteins that form part of the hymenopteran innate immune system and play a role in defence, including killing pathogens, binding to and neutralising endotoxins and modulating immune responses (Brogden, 2005). In honeybees, the most notable AMP is bee defensin-1, which has been identified in both the hemolymph and hypopharyngeal glands and is secreted into honey. Bee defensin-1 has activity against fungi and bacteria through disruption of the cell membrane (Nolan, Harrison & Cox, 2019) and likely plays a role in protecting honey from microbial spoilage (Kwakman et al., 2010). The evolution of AMPs in insects is driven by gene duplication, loss, and divergence along with positive selection by organisms adapting to their unique environments (Bulmer & Crozier, 2004; Viljakainen & Pamilo, 2008). The Camponotus genus differs from other ants in that it lacks the metapleural gland that produces antimicrobial compounds (Schluns & Crozier, 2009), and harbours the endosymbiont Candidatus Blochmannia, which requires the immune system to recognise and tolerate it while simultaneously fighting off other pathogenic microbes (Gupta et al., 2015). This means that honeypot ants are likely to possess unique antimicrobial peptides, distinct both from honey bees and other ant species. Although there have been no studies to date investigating AMPs in C. inflatus, a study on an ant of the same genus, Camponotus floridanus, found unique defensin AMPs, and a hymenoptaecin AMP expressed by genes that are evolutionarily conserved in ants, suggesting the importance of this AMP in the immunity of multiple ant species (Ratzka et al., 2012).

Conclusions

Our research has shown that honeypot ants produce antimicrobial honey with unique species-specific activity that we propose may be linked to their unique evolution and ecology. We tested HPAH against a suite of clinical and environmental pathogens including some commonly used to evaluate therapeutic bee honey, and found that a significant portion of activity likely stems from unique non-peroxide mechanisms. This discovery highlights the potential for the isolation of key compounds or peptides contained within HPAH, which may provide useful leads for therapeutic applications. Our profile of the bacterial and fungal microbiome of Camponotus inflatus demonstrated extreme dominance by single bacterial and fungal species, with additional minor microbial species present that could be linked to foraging behaviour or environmental exposure. These results suggest a potential relationship between microbiota and insect health, which may in turn influence the characteristics of honey. Overall, our study shows that broadening the scope of therapeutic honey research to include other honey-producing hymenopterans can yield valuable insights, and should be encouraged in order to better understand this medically and economically significant commodity.

Supplemental Information

Supplemental Information 1 Honey activity and chemistry raw data.

The raw data for antimicrobial testing, honey chemistry, phenolics and antioxidants.

Click here for additional data file.

The authors would like to thank local Indigenous guides Edie Ulrich, Marjorie Stubbs and Danny Ulrich for their assistance in sourcing the honeypot ants. Thank you also to Yuxin Huo, Kim-Yen Phan-Thien, Svetlana Ryazanova, Iona Gyorgy, Adriana Hoxha, Bridie Stanfield, and Daniel Susantio for their technical assistance.

Additional Information and Declarations

Competing Interests

Author Contributions

Field Study Permissions

DNA Deposition

Data Availability

Dee A. Carter is an Academic Editor for PeerJ.

Andrew Z. Dong conceived and designed the experiments, performed the experiments, analyzed the data, prepared figures and/or tables, authored or reviewed drafts of the article, and approved the final draft.

Nural Cokcetin performed the experiments, analyzed the data, authored or reviewed drafts of the article, and approved the final draft.

Dee A. Carter analyzed the data, authored or reviewed drafts of the article, and approved the final draft.

Kenya E. Fernandes conceived and designed the experiments, performed the experiments, analyzed the data, prepared figures and/or tables, authored or reviewed drafts of the article, and approved the final draft.

The following information was supplied relating to field study approvals (i.e., approving body and any reference numbers):

The sampling of the honeypot ants was performed at Kurnalpi in the Goldfields-Esperance region of Western Australia and we received approval to access the field sites from Honey Ant Tours, a commercial half-day tour where participants are educated on how to dig for honeypot ants. Permission to run these tours has been supplied from the traditional owners of the land.

This land is currently vacant but would otherwise be best described as ‘pastural land’.

The following information was supplied regarding the deposition of DNA sequences:

The data is available at the NCBI: PRJNA957126.

https://www.ncbi.nlm.nih.gov/bioproject/957126

The following information was supplied regarding data availability:

The raw data are available in the Supplemental File.

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
