# Peer review of "Unique antimicrobial activity in honey from the Australian honeypot ant (Camponotus inflatus)"

_PeerJ, doi:10.7717/peerj.15645_

## Round 0.1 · original submission · Minor Revisions

Authors are advised to revise the manuscript according to the reviewers' comments.

·

Basic reporting

I congratulate the authors for selecting an interesting topic, the manuscript is well written, however there are some comments in an annotated pdf which may be taken care by the author before the possible publication of this article.

Experimental design

Satisfactory

Validity of the findings

Satisfactory

Additional comments

Sir,
The comments have been provided in review pdf which may be taken care by the author before the possible publication of this article.

Reviewer 2 ·

Basic reporting

The manuscript “Unique antimicrobial activity in honey from the Australian Honeypot Ant (Camponotus inûatus)” deals with the investigation into the antibacterial properties of honey made by the Australian honeypot ant (Camponotus inûatus) is interesting and valuable. The chemical characteristics and antibacterial activity of honey from honeypot ants (HPAH) against a panel of bacteria and fungi have been thoroughly investigated in this article. The study is very much interesting, and I have really enjoyed the experimental, result and discussion section. However, there is some concern that needs to be addressed before the final decision is made.
Comments
• Ln 31-34: Please rewrite the line.
• In the abstract section, one line on the future impact of the present study can be included.
• LN 43-46: Please expand the concept with more references.
• Ln 96: Please revisit the aims and objective of the study and write in more detail.
• Ln 101: Please provide the latitude and longitude of the place of collection.
• What were the criteria for selecting the strain of bacteria, yeasts, and moulds
• LN 291-299: Please rewrite with better clarity.
• I must appreciate the author for Fig. 5. This figure clarifies the study in a very simple way.

Experimental design

Please see above

Validity of the findings

Please see basic reporting

---

## Round 0.2 · accepted · Accept

The authors have responded satisfactorily to the reviewers' comments. Therefore, the manuscript in its current form can be accepted.

·

Basic reporting

The reviewers have addressed all the comments raised by the reviewers. Thus the article may now accepted for possible publication in Peer J.

Experimental design

Satisfactory

Validity of the findings

Satisfactory

Additional comments

The reviewers have addressed all the comments raised by the reviewers. Thus the article may now accepted for possible publication in Peer J.

Reviewer 2 ·

Basic reporting

The authors made significant changes in the manuscript.

Experimental design

See comments above

Validity of the findings

See comments above

Additional comments

See comments above